# The Factors Influencing Users’ Trust in and Loyalty to Consumer-to-Consumer Secondhand Marketplace Platform

**DOI:** 10.3390/bs13030242

**Published:** 2023-03-09

**Authors:** Yumi Jang, Seongcheol Kim

**Affiliations:** School of Media and Communication, Korea University, Seoul 02841, Republic of Korea

**Keywords:** Karrot, consumer-to-consumer, C2C, secondhand marketplace platform, entrepreneurship, sellers

## Abstract

This study investigates the factors influencing users’ trust in and loyalty to Karrot, a Korean consumer-to-consumer secondhand marketplace platform. This research develops a model with key variables based on the dual model of post-adoption phenomena and adds variables reflecting the specific context of Karrot. An online survey of 305 Karrot users was conducted in South Korea during 19–23 May 2022; the data obtained were analyzed by SEM. The results reveal that two types of trust—trust in Karrot and mutual trust among Karrot users—are direct antecedents of loyalty. Mutual trust among Karrot users is an essential predictor of trust in Karrot. Economic benefits and perceived platform functionality are positively associated with trust in Karrot. Psychological ownership and information interactions were shown to be the important determinants of mutual trust among Karrot users. This study contributes to extending the horizons of post-adoption research by understanding users’ affective and practical motivations for trust and loyalty and by confirming the significant role of two types of trust in forming loyalty. Moreover, this study also provides implications for practitioners of C2C secondhand market platforms to develop their management strategies and expand their customer base.

## 1. Introduction

Consumer-to-consumer (C2C) platforms have prospered under the development of information and communication technology (ICT) and the spread of mobile culture. C2C refers to an online transaction between two private end users to sell or buy items [1]. With the switch toward mobile media in recent years, consumers are encouraged to sell and buy new or secondhand items on shopping platforms without setting up a business or turning to a third party. C2C online transactions are actively taking place all over the world [2]. For example, online C2C e-commerce penetration between 2007 and 2021 in the EU was about 20%. Notably, Asian countries are leading this sector. For instance, the C2C market accounted for almost a quarter of China’s online retail sales in 2022 [2]. In Korea, secondhand transactions through C2C platforms have been invigorated in recent years; these online marketplaces attract attention as a new trend in the shopping app market, reaching 19.28 million users in August 2022 [3]. In particular, the MZ generation—a Korean term referring to people who were born between 1980 and 1994 (millennials) and between 1995 and 2004 (Generation Z)—occupies around 60% of secondhand marketplace platform users. According to the survey by GoodRich, a Korean insurance management agency, 83% people in their 20s and 30s bought or sold secondhand items in 2020 [4]. The reason for this captivation is twofold. First, a consumption trend that values use and experience rather than ownership has been encouraged [4]. Second, high-priced luxury secondhand trading among people, especially the MZ generation, has been activated [4,5]. Accordingly, the secondhand market has steadily grown and expanded. The size of Korea’s secondhand market is estimated to be about USD 15.43 billion in 2021, which is up dramatically from USD 3.09 billion in 2008 [4].

In South Korea, three representative secondhand marketplace platforms—Karrot, Joonggonara, and Bungaejangter—dominate the market. Among these big three, the hyperlocal e-commerce app Karrot has made remarkably rapid strides, having 93% of the market share, surpassing the previous long-standing market leader Joonggonara in terms of market share within a short period after expanding service coverage nationwide in 2018 [6]. ‘Karrot’ is an abbreviation for “the market in your neighborhood” in Korean; as its name indicates, it facilitates users’ buying and selling of used items in their local areas. Trading used items via Karrot is available only among local residents living in a radius between 4 km and 6 km. Most users meet with their counterparts in person in the same neighborhood without sharing their address or telephone number.

In-person trade among neighbors has reduced the number of fraudulent cases routinely found in the online trade of used goods [6]. Historically, online secondhand transactions have had a high risk of fraud because the other party cannot necessarily be trusted [6]. In contrast, Karrot achieved its rapid growth by securing ‘trust,’ thus overcoming the biggest problem of the existing secondhand market platforms [6]. In other words, Karrot’s soaring success is based on an entrepreneurial approach that combines a hyperlocal service targeting a small neighborhood and a direct transaction method. This new attempt satisfied online secondhand platform users’ appetite and eventually generated user trust in Karrot. Nevertheless, Karrot is not a purely omnipotent secondhand marketplace app; it also experiences weaknesses associated with trust. Growing pains, such as no-shows and poor customer service, negatively influence user trust in Karrot.

Korea is an ICT powerhouse, and around 97.4% of Korean people are using smartphones as of 2023 [7]. Due to advanced mobile culture, mobile C2C secondhand marketplace platforms are vigorously developing and attracting users’ attention in Korea. In particular, Karrot is the leading secondhand marketplace platform with 93% of the market share. Its monthly active users are 16.45 million [6]. Karrot was launched in 2015 as a late mover in the market, but it has quickly become a consumer favorite [3,8]. This study focused on Karrot, a Korean case, because it is a game changer in the secondhand market. Despite Karrot’s continuous growth and attention in the market, however, there is a lack of research on the topic, and studies on the trust of Karrot users are particularly hard to find. Thus, the purpose of this paper is to investigate what factors make users trust and continuously use Karrot, a Korean representative C2C secondhand marketplace platform. This study focuses on two different types of trust: (1) mutual trust among users and (2) trust in the platform provider. Simultaneously, their influence on loyalty to Karrot is also investigated, anchoring its theoretical basis on the dual model of post-adoption phenomena. Furthermore, this study also introduces variables reflecting Karrot’s specific context. Unlike previous studies on C2C secondhand marketplace platforms, which adopt platform-specific variables as one dimension, this study divided Karrot-specific variables into a constraint-based mechanism and dedication-based mechanism to better reflect the characteristics of the C2C secondhand marketplace platform. Constraint-based variables are economic benefit and platform functionality, while dedication-based variables are psychological ownership and social interaction. Bilateral variables, which belong to both mechanisms, such as no-shows were employed in this study. This approach would be more helpful to lead to optimal results in the context of online second-hand transactions. Consequently, this study is expected to contribute to extending the horizons of post-adoption research by understanding users’ various motivations for a different type of trust and loyalty and providing more practical guidelines for practitioners of the C2C secondhand market platform.

## 2. Research Background

### 2.1. C2C Secondhand Marketplace Platform

Trading secondhand items online is by no means a new concept. However, by combining a convenient mobile app, hyperlocal community functionalities, and the receptivity of secondhand products, consumers, especially the MZ generation, actively participate in transactions on a reliable C2C secondhand marketplace platform [9]. Unlike firsthand products that allow consumers to check the quality of the goods with his/her own eyes and purchase them, secondhand products lack evaluation standards applicable to their quality and maintenance. This means that the traded secondhand item has a relatively high risk and uncertainty for the transaction. Therefore, trust, as a prerequisite of a successful transaction, plays a more critical role in online second-hand marketplace platforms than in ordinary e-commerce platforms [10,11]. For example, Luo [12] examined that e-commerce service quality including system quality, security assurance, product variety and service support, and community quality had direct and interacting effects on users’ perceived trust, which consequently affected their transaction intention in the context of a Chinese secondhand marketplace platform. Another prior study [13] discovered that trust and engagement with the platform had a significant positive effect on consumers’ intention to re-use a C2C secondhand marketplace platform.

Karrot ushered in a new era in Korea’s online secondhand marketplaces. Karrot is a community-based service app published in 2015 to buy and sell with neighbors. In South Korea, Joonggo Nara—a massive online community on the Korean web portal Naver—was the pioneer that has powered the steady growth of online sales of pre-owned goods since its foundation in 2003. However, with the debut of Karrot in 2015 and its expanded service coverage throughout the nation since 2018, the competitive landscape was reshaped.

Karrot stands for “the market in your neighborhood” in Korean. As its name implies, it facilitates users’ buying and selling of used items in their local places. This is a unique feature of Karrot that only displays an item list of sellers located within a radius between 4 km and 6 km. Real-time transfers and confirmations between users are possible through the ‘Karrot chat’ function. Most users meet with their counterparts in person in the same neighborhood without separately sharing their addresses or telephone numbers. With Karrot’s own payment system ‘Karrot pay,’ users no longer need to exchange personal information such as account numbers and account holders between users on the street, run a separate banking app to verify transfer details, or even prepare cash for transactions. Karrot’s easy-access process is another key characteristic of its noticeable expansion in the secondhand platform market. The sign-up process requires only a user’s location and contact information, without any complicated verification process. The unique combination of a lower entry barrier, privacy protection, and in-person trade among neighbors has reduced the number of fraudulent cases routinely found in the online trade of used goods [6]. In addition, by encouraging users to give away sundry items on the app that will otherwise be thrown away, asking others to take them free of charge, Karrot enables a form of community-based recycling [6].

Besides its used item trading, Karrot provides various social interactive functions such as the ‘manner meter’ (an indication of whether the seller is friendly or an overall good seller), ‘Karrot chat’ (one-on-one chat between users), ‘my local’ (SNS), and ‘nearby’ (bulletin board), where users share information and events in their neighborhood and help each other. These interactions develop a sense of community fellowship between users [14]. Based on this distinctive mix of various features, Karrot is now deemed a front-runner in the new business category of hyperlocal e-commerce and local community apps in Korea [6].

According to Mobileindex [15], the number of people using secondhand marketplace apps has jumped 141% year-on-year to 16.4 million as of April 2020. Of this total, Karrot’s share reached 93%, holding an absolute grip on the rapidly growing sector. Karrot has become one of the most downloaded apps among Korean users, with more than 22 million downloads, outpacing Netflix, Instagram, and TikTok [15]. Karrot’s registered users stand at 21 million, or one user per Korean household across the nation as of September 2021 [6]. Its monthly active users have jumped nearly 30-fold in the past three years to 14.2 million as of January 2021 [16]. Industry estimates suggest that Karrot executed mobile transactions worth USD 840 million in 2020 [6]. Indeed, “doing Karrot” is a term widely accepted for selling or buying used goods via the app among Korean people. Buoyed by its soaring popularity, Karrot has raised a total of USD 205 million in a series of funding, with its enterprise value set at USD 2.7 billion [16,17]. Accordingly, Karrot became the sixteenth Korean ‘unicorn,’ or privately-owned company with a valuation of over USD 1 billion, in August 2021, according to the Ministry of SMEs and Startups of South Korea [17]. In recent years, Karrot has expanded its operations internationally and reached a broader market, including the UK, US, Canada, and Japan.

However, not everything is going smoothly with this hyperlocal community app. As Karrot relies on individual transactions, some disputes cannot be avoided. The issue is the lack of regulatory and policy systems to manage a growing number of complaints. According to data filed to the National Assembly in September 2021, the number of dispute arbitration applications involving Karrot during the January–August period in 2021 was 1167, more than a 60-fold jump from just 19 in 2019 [6]. Another issue is trade of luxury items through Karrot; there are a considerable number of users who use Karrot to do arbitrage on big-ticket items such as luxury handbags worth thousands of dollars for the sake of profit [5]. However, as some luxury business operators or individuals in the high-income class reportedly exploit the regulatory loopholes related to such platforms to avoid taxes, there is rising criticism of a lack of a system to prevent or clamp down on such illegal transactions [6].

### 2.2. Dual Model of Post-Adoption Phenomena

IS studies have insisted that consumers’ post-adoption behaviors such as continuance use, repeated use, word-of-mouth, and willingness to pay are critical factors for a firm’s success in a highly competitive marketplace [18,19]. In this sense, users’ patronage at the post-adoption stage is highlighted as a key to the survival of online service providers [18]. In the IS literature, users’ continuance behavior has been explained by user satisfaction and commitment [20]. Gradually, the suggestion that user commitment is more critical for predicting a user’s volitional and continued use of the service has been acknowledged in that commitment is more action-oriented and resilient to other situational influences than satisfaction [20].

Commitment is defined as a “psychological state that compels an individual toward a course of action” [21] (p. 303). Given that commitment differs conceptually from satisfaction and is regarded as a significant direct and indirect predictor of loyalty or intention for continued use [22], the dedication–constraint framework of commitment has been leveraged to investigate online service users’ post-adoption behavior [18,20,23,24]. Rooted in social exchange theory, the dedication–constraint framework suggests that user loyalty results from two mechanisms: personal dedication and constraints that underlie an individual’s development of commitment [20,24].

Personal dedication indicates an individual’s desire to maintain a relationship, which focuses on the prospect of long-term mutual benefits. According to prior research, dedication-based mechanisms are derived from affective commitment [25]. IS researchers explained that affective commitment means positive regard for and desire-based attachment to a service provider, and this encourages a user to maintain their long-term relationship [20,25]. They emphasized that because the development of affective commitment is attributed to users’ immersive experience, which fulfills their psychological needs for comfort and confidence, affective commitment reflects one’s involvement in and belonging to the relationship with the service (provider). Prior research also highlighted that dedication-based mechanisms enforce user loyalty or post-adoption behaviors [18]. However, the dedication perspective alone cannot fully capture the user’s post-adoption decision-making processes [23]. To more comprehensively understand the reason that users maintain and retain relationships with a service provider, it is necessary to first comprehend both dedication- and constraint-based mechanisms [24].

Constraint indicates the “forces that constrain individuals to maintain relationships regardless of their personal dedication to them” [26] (pp. 595–596). Prior studies explained that constraint-based mechanisms are deeply related to the concept of calculative commitment [20]. Calculative commitment refers to the degree to which individuals perceive that they are locked into the relationship with their current service provider because of the potential costs of switching to an alternative [27,28]. Many researchers consider calculative commitment an important predictor of loyalty to online services [28,29]. For instance, based on the side-bet theory, some studies explained that calculative commitment forces a user to reduce interest in alternative services and maintain the relationship with the service provider [24,27]. Kim [24] insisted that constraint-based mechanisms stem from service-specific investments, and constraint factors play an essential role in maintaining the relationship with the current service provider because service users need to spend a significant amount of time learning what the service is and how to use it. Allen and Meyer [27] confirmed that high withdrawal costs have an effect of binding a user to continued use of their current service.

Using these two mechanisms, Kim and Son [18] proposed a dedication–constraint dual model that explains post-adoption behaviors in the context of online services. This model uses loyalty and switching costs to capture dedication- and constraint-based mechanisms, and it introduces perceived benefits and service-specific investments as the antecedent of those two commitment mechanisms. Zhou et al. [20] adapted the dedication–constraint framework of commitment and developed a model of social-virtual-world service users’ continuance intention. Kim [24] deployed an integrative framework of a dedication- and constraint-based model to examine user loyalty toward mobile messenger applications. The research by Lin et al. [30] and Kim [24] elaborated that dedication- and constraint-based mechanisms can explain considerable variance in customer loyalty. In particular, they emphasized that constraint factors significantly influence customer loyalty more than satisfaction.

To extend the horizons of post-adoption behavior research, the current study develops and tests a model that explains post-adoption behaviors in the context of Karrot. Drawing on a dual model of relationship maintenance in consumer behavior research [23], this study proposes a conceptual framework to investigate customer behavior in the C2C secondhand market platform. In particular, based on prior studies, our model predicts that customers’ post-adoption reactions to the platform are driven primarily by two contrasting mechanisms: (1) the dedication to the platform as generated by the expectation for a long-term mutual relationship and (2) the constraint that makes it difficult for users to switch to an alternative. Under the assumption that dedication- and constraint-based mechanisms may establish user loyalty based on prior studies, these two mechanisms are centered on the concept of loyalty in this study.

### 2.3. Two Types of Trust

Trust is defined as “the willingness of a party to be vulnerable to the actions of another party based on the expectation that the other will perform a particular action important to the trustor, irrespective of the ability to monitor or control that other part” [31] (p. 712). In terms of relationships, trust plays an essential role: it enhances the value of and increases capabilities in the relationship [32,33]. It also positively impacts relationship performance [34]. In other words, trust works as both a lubricant [35] and glue [36] in the relationship. As C2C platform users encounter two distinct entities—other users and the C2C platform provider—this study considers two categories of trust relationships: mutual trust among users and user trust in the platform provider [37].

Mutual trust among users refers to “a perception that users of a community are willing to be vulnerable to the actions of other users in a business transaction based on past experience” [37] (p. 150). To put it more simply, it indicates a shared trust relationship among users. As antecedents of mutual trust, social interactions which indicate that two or more users are mutually oriented toward each other were presented in a prior study [38]. According to a previous study, mutual trust among users is a credible predictor of loyalty to a C2C platform [37].

Trust in the platform provider refers to a general belief that the platform provider is trustworthy [39] and encompasses the impression of the integrity, benevolence, and ability of the platform provider [40]. Constraint-based factors related to economic exchanges and cost and benefit (e.g., the quality and quantity of goods to be exchanged and the price [41]) affect trust in the platform provider [40]. It has been discovered that this trust is one of the most significant predictors of loyalty [37,42]. Additionally, prior studies demonstrated that trust in the platform provider is closely correlated with mutual trust among platform users. Chen et al. [37] explained that trust in the platform provider benefits from mutual trust among platform users, and Zucker [43] insisted that mutual trust among users can be transferred to institutional trust. Due to the nature of Karrot, where interactions between the platform and other users simultaneously occur, this study assumed that both trust toward Karrot and trust among other Karrot users should be considered. Therefore, the following hypotheses are proposed.

**Hypothesis 1.** *Mutual trust among Karrot users is positively related to trust in Karrot*.

**Hypothesis 2.** *Mutual trust among Karrot users is positively related to loyalty*.

**Hypothesis 3.** *Trust in Karrot is positively related to loyalty*.

### 2.4. Platform-Specific Factors of Karrot

The constraint-based mechanism is associated with the economic exchange and service performance [41]. In the context of Karrot, this study considers economic benefit and platform functionality as constraint-related factors. These factors can exert an influence on users’ trust in Karrot.

So far, previous studies on secondhand trading have focused on the economic benefits in terms of cost savings [44]. However, there is a prominent tendency to raise personal profits among those who participate in Karrot for the entrepreneurial purpose of selling unnecessary items [45]. On Karrot, users can sell a wide variety of items such as large furniture, living items, and food that is difficult to transact on general secondhand marketplace platforms [14]. Thus, sellers on Karrot feel relatively free from organizing sale items, get more opportunities to make a deal with a buyer, and, consequently, make profits. The increased product variety works as a source of users’ welfare gains [46], and this extended freedom of selling can improve the seller’s surplus gains and welfare [46]. According to Ahn [45], quite a few users earn a high income by selling expensive items such as luxury goods worth multiple thousands of dollars or selling large quantities of goods. As Karrot provides intuitive functions which make transactions quick and straightforward, and because the goods are directly sold within the neighborhood, users can gain higher profits in a short time if they have a good strategy and work hard [45]. These behaviors of sellers can be described as entrepreneurial. Thus, this study focused on economic benefits through generating additional income and defined economic benefit as financial income that users earn from trading secondhand goods. Economic benefits could be one of the “desirable consequences” of secondhand trading [44,47] (p. 61), and it might be considered the kind of relationship benefit that Palmatier et al. [45] described. According to Palmatier et al. [48], relationship benefits indicate various functional and social benefits and rewards that increase customers’ personal value. They also indicated that these benefits are connected to trust. Moreover, Hosmer [49] speculated that trust could be established when a firm meets the needs of a service user in economic exchange. Based on prior studies, it can be assumed that users who gain economic benefits through secondhand deals in Karrot have trust in the platform. Therefore, the following hypothesis is proposed.

**Hypothesis 4.** *Users’ economic benefits are positively related to trust in Karrot*.

Platform functionality refers to a system’s capability, which provides users with what they want to meet their objectives [50]. In a lot of IS research, platform functionality has been considered as perceived usefulness, and it influences users’ attitudes [49,51]. Young and Benamati [52] argued that platform functionality includes informational use, transactional use, and customer service use. In an online shopping environment, platform functionality means the support for platforms’ core products and services so that they can enhance transactions and help users achieve their shopping goals [51]. Turban et al. [53] highlighted that a C2C platform must establish a well-functioned-system capability to smoothly assist users’ service use by providing content and tools for successful product searching, selling, and buying. However, Karrot is relatively weak in providing satisfactory platform functionality, especially in customer service use. For instance, Karrot does not operate a customer call center and receives users’ complaints only in writing [54]. In addition, even if a seller’s sales post is processed blindly due to multiple reports by other users, there is no separate notification to inform unless a seller checks it in a roundabout way. This operational policy of Karrot may adversely affect its users’ perception of its platform functionality. To understand the user’s perception of Karrot’s platform functionality associated with customer support and its influence on trust in the platform, this study suggests the following hypothesis.

**Hypothesis 5.** *Perceived platform functionality is positively related to trust in Karrot*.

Prior studies found that a dedication-based mechanism is associated with users’ psychological needs for comfort and confidence [20,25]. In the context of Karrot, perceived mutual benefits-related factors such as psychological ownership and social interactions were proposed to understand the dedication-based mechanism. These factors can exert an influence on the mutual trust among Karrot users.

Psychological ownership refers to a state in which an individual perceives that the target of ownership is theirs [55]. To protect the target of ownership, individuals feel a sense of responsibility and shared interests with other owners [56]. With the concept of ‘with my neighbor,’ Karrot requires users to verify their residential area, and it assures users that other users they meet in Karrot are their actual next-door neighbors. Moreover, given that 93.3% of Karrot users are both buyers and sellers [57], it can be assumed that Karrot users are actively connected. Indeed, Karrot has become a true peer-to-peer secondhand marketplace as well as a local living community platform. As users feel a sense of belonging, commonality, affection, and companionship through Karrot activities, users share significant psychological ownership toward Karrot with other users. According to Palmatier et al. [48], when people perceive that they have similar lifestyles, cultures, values, or goals with other people, they feel trust toward them. In Karrot, users live in an area that shares the same culture or atmosphere. In addition, a user meets other users who have a similar way of evaluating the value of goods as well as similar goals they want to achieve through secondhand transactions. Accordingly, users have psychological ownership in Karrot and feel trust toward other users who share similar senses and interests. Based on this discussion, this study therefore proposes the following hypothesis.

**Hypothesis 6.** *Psychological ownership is positively related to mutual trust among Karrot users*.

According to Huemer [33], interacting among users and the way that activity occurs is fundamental for developing the platform market. Chen et al. [37] also insisted that interactions among users is the most important activity in C2C e-commerce platforms. Cheung et al. [58] found that interaction between consumers is influential in evoking consumers’ cognitive and emotional engagement and consequently impacts behavioral intention. Abdul-Hgani et al. [59] indicated that consumers interact on C2C platforms to socialize, exchange information, and trade. Emotional and information interactions are representative categories of social interactions, with emotional interaction referring to “the interaction of affects, moods, and emotions among users” and information interaction indicating that users share their information and knowledge with others [37] (p.152). Chen et al.’s study [37] demonstrated that information interaction includes various activities related to information and knowledge-seeking, provision, exchange, and sharing, and these activities then generate more emotional interactions among users. In Karrot, users are connected as sellers and buyers in the same local community and interact emotionally and informatively with each other through communicative functions [14] such as ‘manner meter’ (an indication of whether the seller is friendly or an overall good seller), ‘Karrot chat’ (one-on-one chat),’ ‘my local’ (SNS), ‘nearby’ (bulletin board), and others. For instance, Karrot users exchange information such as local information, event information, life tips, or how to solve problems when they arise through a bulletin board or SNS function in Karrot. Regarding secondhand item trading, they collect information such as how to avoid secondhand sales scams or tips for negotiating prices. Meanwhile, through those functions, users exchange emotional empathy such as cheering, support, and consolation with other users. This means that users obtain values from various interactions in Karrot, and these values could ultimately be connected to their attitude and post-adoption reaction [59]. Palmatier et al. [48] clarified that frequent and good-quality communications and interactions are crucial in relationships, and these emotional and informational exchanges positively influence trust. Chen et al. [37] also discovered that emotional and information interactions are key predictors of mutual trust among users. Given that various information and emotional interactions between users are significant features of Karrot, this study proposed following hypotheses.

**Hypothesis 7.** *Information interactions are positively related to mutual trust among Karrot users*.

**Hypothesis 8.** *Emotional interactions are positively related to mutual trust among Karrot users*.

This study also presented perceived risk-related factor such as no-shows. This bilateral factor reflects both affective and practical motivations of Karrot users.

No-shows refer to a situation where the customer who has already made a reservation does not appear at the appointment time without prior notification of changes or cancellations [60]. With the increase in users, the no-show problem has been on the rise in Karrot [61]. Due to Karrot’s characteristics of distinctive and cheaper goods and free gifts, there is a considerable possibility that users who make a transaction might treat a reservation without deep consideration. Buyer no-shows can cause problems that deprive other user the opportunity to get wanted or needed items. For sellers, the occurrence of no-show behavior without prior notice can create a lot of confusion, difficulty, and additional work [62]. Palmatier et al. [48] demonstrated that relational investments such as time, effort, spending, and resources are crucial antecedents of trust. If such an investment from a seller suffers from a buyer’s no-show behavior, it can cause negative emotions or attitudes among sellers and severe consequences of lost trust among users and, in turn, toward the platform [43]. The following hypotheses are therefore proposed.

**Hypothesis 9.** *No-shows are negatively related to trust in Karrot*.

**Hypothesis 10.** *No-shows are negatively related to mutual trust among Karrot users*.

## 3. Methodology

### 3.1. Data Collection and Analysis

Data were collected through online surveys for Korean samples from 19–23 May 2022. Korea is one of the leading countries in mobile services, and the C2C secondhand marketplace platforms represented by ‘Karrot’ are perceived to have significant growth potential. Thus, we recruited participants with experience in secondhand item selling and social network services in Karrot through an online survey company. In the online questionnaire, participants filled out questions about demographics and their secondhand trading and social interactions on Karrot. The survey instrument was constructed based on established measures of constructs from marketing and IS literature, which was adapted to be applicable to the context of our proposed model. All items were anchored on a seven-point Likert scale ranging from “1 = strongly disagree” to “7 = strongly agree.” Table A1 in Appendix A shows all items used for the survey. This study adopted structural equation modeling (SEM) using AMOS to analyze the theoretical propositions, which was chosen for its efficiency in simultaneously testing multi-staged causal relationships [63].

### 3.2. Sample Characteristics

Table 1 shows the following features: main shopping channel, number of secondhand marketplace platforms in use, most used secondhand marketplace platform, age, gender, occupation, education, and region. Respondents spent an average of KRW 315,119 (USD 240.92) per month on shopping. Respondents mostly used the online shopping website and mobile shopping app for shopping. Interestingly, more than 10% of respondents used the mobile secondhand marketplace app as their main shopping channel, showing the growth of the secondhand market in the broader consumption market. Respondents used 1.94 secondhand marketplace platforms on average, and most of the respondents (92.5%) used Karrot the most for secondhand trading. Regarding Karrot, respondents used Karrot for 28 months on average. They accessed Karrot an average of 5.7 times per week and traded used items an average of 2.81 times per month on Karrot. Respondents spend an average of KRW 57,360 (USD 43.85) per month on buying used items and earn an average of KRW 42,354 (USD 32.38) per month selling used items in Karrot. This means that respondents spent about 18.20% and earned approximately 13.44% of total shopping spending on Karrot.

### 3.3. Test of Measurement Model

The reliability test, which examines internal consistency within a construct, was performed by Cronbach’s alpha and composite reliability (CR). As shown in Table 2, all constructs show a value above the threshold of 0.7 for both Cronbach’s alpha and CR, as adopted by Werts et al. [64]. Convergent validity reflects the extent to which the indicators of a construct are more strongly correlated to each other than to indicators of other constructs. To test convergent validity, we examined CR, factor loading, and average variance extracted (AVE). It is acceptable for an individual item factor loading to be greater than 0.5, for CR to exceed 0.7, and for AVE to exceed 0.5 [65]. The factor loadings of all observed variables or items usually range from 0.661 to 0.926. All other values were above the marginal standard, as shown in Table 2. Thus, the convergent validity of the construct was adequate. To test discriminant validity, which shows that measurement items load highly on their theoretically assigned constructs and do not load on other factors, this study examined the table correlation of constructs and latent square root of AVE. To satisfy discriminant validity, the square root of AVE should be greater than the correlations between different constructs [66]. As Table A2 in Appendix A presents, the square root of AVE for each construct in this study exceeded the correlations between the construct and other constructs. Thus, discriminant validity was established.

## 4. Results

In this section, we attempt to verify our hypotheses using SEM analysis. Our research model could explain 70.2% of the variance in loyalty, 72.0% of the variance in trust in Karrot, and 29.4% of the variance in mutual trust among Karrot users. Table 3 and Figure 1 show our research model with a summary of the results following hypothesis testing; a boot-strapping procedure was used to confirm the significance of the path coefficients. Based on the analysis, six out of the eleven hypotheses were supported. First, regarding trust, mutual trust among Karrot users was significantly related to trust in Karrot (H1 was supported), but it showed the opposite direction for loyalty, despite a significant relationship (H2 was not supported). Trust in Karrot was significantly related to loyalty (H3 was supported). Among service-specific benefits, economic benefits and perceived platform functionality were significantly related to trust in Karrot (H4 and H5 were supported). Among perceived mutual benefits, psychological ownership and information interactions had a significant relationship with mutual trust among Karrot users (H6 and H7 were supported). Emotional interactions were not significant (H8 was not supported). As perceived risk, no-shows did not have a significant relationship with either trust in Karrot or mutual trust among Karrot users (H9 and H10 were not supported).

## 5. Discussion and Conclusions

### 5.1. Key Findings and Implications

The purpose of this study was to investigate the motivations for different types of trust and their influence on loyalty in the context of Karrot. Thus, this study hypothesized and tested key variables reflecting the dual model of post-adoption phenomena and platform-specific factors of Karrot, a leading Korean C2C secondhand marketplace platform. Several findings can be derived from this study.

First, this study confirms the significant role of two types of trust in developing loyalty while extending the scope of trust research to the two-sided mobile platform market. More specifically, mutual trust among Karrot users had a positive relationship with trust in Karrot, and trust in Karrot positively influenced loyalty. Additionally, the paper shows that mutual trust among Karrot users was an antecedent of loyalty, but it was contrary to our prior expectations. Initially, we predicted that higher mutual trust among Karrot users would be a significant factor in increasing the loyalty of Karrot users because they would use a platform where trustworthy people gather. However, the result was the opposite. A plausible reason is that mutual trust among users can be meaningful only under the condition that the platform’s performance—entailing the service function and customer support—are satisfactory, and thus trust in the platform is established. Otherwise, it can have the opposite effect. This signifies that Karrot users recognize trust in Karrot and mutual trust among Karrot users as different dimensions. They consider that trust in the platform is more vital for maintaining the relationship than mutual trust among users. Practically, this result implies that because mutual trust among users itself is not enough to lead to users’ positive post-adoption reactions, the role of the platform is essential. Although platform users trust each other, their post-adoption reaction to the platform would not always be positive. When the platform plays a mediating role in increasing users’ welfare and decreasing the risks, and when users think that the platform is trustworthy (and therefore demonstrate trust toward the platform), they are likely to maintain the relationship with the platform. Accordingly, it is vital for the platform provider to make the effort to prevent negative events or perceptions related to the platform and put more weight on making the distinguished value of the platform itself.

Second, as hypothesized, the results support the idea that economic benefit is one of the key determinants of increasing the trust in Karrot. The findings support the argument that economic benefits could be one of the “desirable consequences” of secondhand trading [44,47] (p. 61). Overall, the study found that Karrot users put the highest weight on the economic benefit of their trust in Karrot. This result implies that providing sufficient opportunities for generating profit is the most important responsibility of Karrot. For example, it may be possible to consider ways to diversify the types of items that can be sold, to provide delivery helper service, to hold special venues or events for secondhand trading, or to increase the convenience of payment methods.

Perceived platform functionality also was shown to be an antecedent of trust in Karrot. This means that users place trust in the platform when the platform functionality is well-equipped and operating smoothly. Given that developing the characteristic value of the platform is important for developing trust and ultimately loyalty to the platform, as we explained earlier, Karrot should not shirk its responsibility for organizing, developing, and managing its platform functionality.

Third, psychological ownership and information interactions were shown to be antecedents of mutual trust among Karrot users. Information interactions showed the largest effect size for the mutual trust level of Karrot users. Cheung et al. [67] discovered that emotional and information interactions are important drivers of consumers’ perceived value based on the SOR framework. Another prior study identified that interactions between consumers have an impact on evoking emotional attachment and behavioral intentions grounded on service-dominant logic [58]. According to Palmatier et al. [48], people trust each other when they perceive that they have similar lifestyles, cultures, values, or goals. They also indicated that frequent and good-quality communication and interactions have a positive influence on that trust. Our results partially confirmed the previous studies’ arguments. The findings of this study showed that information interactions are key predictors of mutual trust among platform users [37]. Nowadays, Karrot has become a substantial neighborhood life platform. People not only trade secondhand items but also have a social life on Karrot. For instance, Karrot users share information about their town, life, real estate, and recruiting, and they look for friends on the block through Karrot. Since people are more likely to get information to make progress on the situation, it seems that cognitive engagement rather than emotional engagement exerts a stronger influence on establishing mutual trust among users. However, it is also obvious that local residents gather and form various relationships and share emotional empathy around Karrot. Both informational and emotional interactions are great assets for Karrot. A previous study highlighted that emotional and information interactions with the mediating role of entrepreneurship have a significant influence on the internationalization of digital startups [68]. Therefore, to leverage these assets and expand the business abroad, Karrot should develop and nurture a place where users can continuously feel psychological ownership and actively interact with each other based on entrepreneurship.

As one of the new attempts to study the secondhand marketplace platform Karrot, this research provides empirical evidence examining how Karrot works and the main benefits for users, especially sellers. Karrot has achieved rapid growth, making direct secondhand trading in the neighborhood easy and convenient. However, the variety of attractive items posted by sellers and the increase in sellers’ profit from selling these items also played a major role in Karrot’s growth. This study is meaningful in that it identifies the role and influence of these factors. With deliberate consideration of their relationship with users, Karrot will be able to maximize its synergies with users and pursue sustainable growth.

This study makes several contributions. Academically, this study attempted to extend the horizons of post-adoption research by developing and testing a model that explains post-adoption behaviors in the context of Karrot. More specifically, this study tried to understand users’ affective and practical motivations for trust and loyalty in Karrot by focusing on a dual model of relationship maintenance which consists of two contrasting mechanisms—1) the constraint-based mechanism and 2) the dedication-based mechanism—and two types of trust—1) trust in the platform and 2) mutual trust among users. The findings provided empirical evidence for understanding how Karrot works and maintains a long-term relationship with its users. Practically, the findings could be a reference for practitioners to develop their management strategies and expand their customer base. In detail, this study provides implications for practitioners on what value they need to provide to drive user loyalty. In addition, this study may give clues for Karrot to plan strategies to attract more users and strengthen its leading position. For instance, there are people who participate in secondhand deals for the entrepreneurial purpose of selling unnecessary items to raise personal income. If Karrot develops a special marketing strategy that promotes their entrepreneurial trading activities, it can attract more general users.

### 5.2. Limitation and Future Research

Several limitations and suggestions for future research should be noted. First, participants were recruited from one country. Karrot has expanded its overseas market, including the UK, US, Canada, and Japan. Thus, results will likely vary by country due to differences in social and cultural environments. For instance, because the scale that defines the ‘neighborhood’ and the way of life may be different overseas than in Korea, users in other countries could have different perceptions and preferences for the benefits or risks of Karrot. If future research considers these differences and compares them, more prosperous and reliable results could be obtained. Moreover, this study focused on significant characteristics of Karrot as initial exploratory research. Further analysis can deal with more detailed features and differences from other secondhand marketplace platforms, such as trade and payment methods, security protection, and compensation systems for fraud.

Notwithstanding its limitations, this study provides an expansion of the literature on secondhand marketplace platforms by examining the effects of factors reflecting the specific context of Karrot and two different types of trust. Researchers and practitioners must begin to recognize how users’ perceptions of these factors affects their loyalty to a secondhand marketplace platform. Notably, other secondhand trading platforms such as Joonggonara and Bungaejangter have implemented various measures and policies to attract more users in an attempt to hold Karrot in check. Considering that Karrot users’ primary interests are economic benefits and platform functionality, the relationship could be different under a more detailed observation. We hope that our study will ignite further research on secondhand marketplace platforms and consumer behavior.

## Figures and Tables

**Figure 1 behavsci-13-00242-f001:**
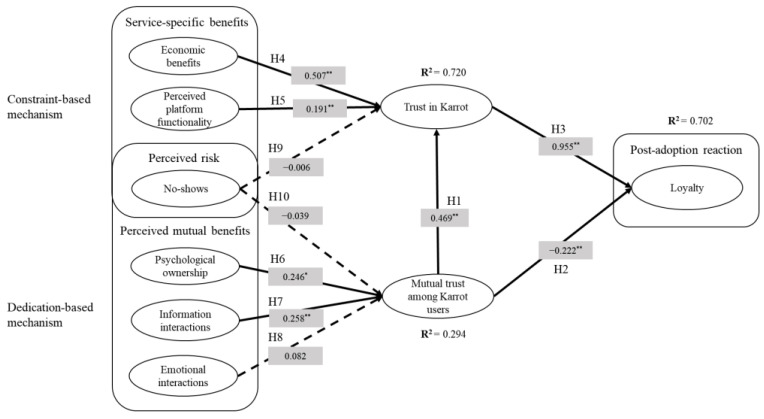
Research model and result; * *p* < 0.05, ** *p* < 0.01.

**Table 1 behavsci-13-00242-t001:** Sample Characteristics (*N* = 305).

Characteristics	Frequency	Valid Percent
Main shopping channel		
Online shopping website	124	40.7
Mobile shopping app	98	32.1
Mobile secondhand marketplace app	32	10.5
Offline store	30	9.8
Brand official web site	8	2.6
Online secondhand marketplace web site	7	2.3
Brand official mobile app	3	1.0
TV home shopping	3	1.0
Number of secondhand marketplace platform in use		
1	97	31.8
2	151	49.5
3	36	11.8
More than 4	21	6.9
Most used secondhand marketplace platform		
Karrot	282	92.5
Joonggonara	10	3.3
Bungaejangter	8	2.6
Local community	5	1.6
*Age:*		
20s	75	24.6
30s	76	24.9
40s	79	25.9
50s	75	24.6
Gender		
Male	151	49.5
Female	154	50.5
Occupation		
Student	32	10.5
Housewife	39	12.8
Office worker	155	50.8
Professional	32	10.5
Self-employed	22	7.2
Other	25	8.2
Education		
Middle school	1	0.33
High school	34	11.15
College	239	78.36
Advanced degree	31	10.16
Region		
Seoul	104	34.1
Busan	12	3.9
Daegu	18	5.9
Incheon	15	4.9
Gwangju	7	2.3
Daejeon	13	4.3
Ulsan	4	1.3
Gyeongi-do	77	25.2
Gangwon-do	11	3.6
Chungcheongbuk-do	44	14.4

**Table 2 behavsci-13-00242-t002:** Descriptive Statistics of Variables.

Variable Name	Code	No of Items	Mean (Std. Dev)	Cronbach’s Alpha	AVE	Composite Reliability
Economic benefits	EB	4	5.66 (0.99)	0.915	0.730	0.915
Perceived platform functionality	PF	4	4.59 (1.02)	0.944	0.808	0.944
Psychological ownership	PO	4	4.29 (1.16)	0.880	0.651	0.880
Information interactions	II	4	4.53 (1.18)	0.915	0.731	0.916
Emotional interactions	EI	4	3.73 (1.49)	0.932	0.781	0.934
No-shows	NS	4	5.10 (1.13)	0.942	0.805	0.943
Trust in Karrot	TK	4	5.10 (0.88)	0.901	0.701	0.904
Mutual trust among Karrot users	MT	4	4.62 (0.94)	0.887	0.669	0.889
Loyalty	LO	4	5.58 (1.01)	0.939	0.800	0.941
Total items		36				

**Table 3 behavsci-13-00242-t003:** Direct Impact of Model: Standardized Regression Weights.

H	Relations			Std. Estimate	S.E.	C.R.	*p*-Value
H1	MT	→	TK	0.469	0.051	8.243	0.000
H2	MT	→	LO	−0.222	0.078	−3.330	0.000
H3	TK	→	LO	0.955	0.106	11.811	0.000
H4	EB	→	TK	0.507	0.049	9.134	0.000
H5	PF	→	TK	0.191	0.038	3.790	0.000
H6	PO	→	MT	0.246	0.084	2.035	0.042
H7	II	→	MT	0.258	0.073	2.600	0.009
H8	EI	→	MT	0.082	0.064	0.799	0.424
H9	NS	→	TK	−0.006	0.031	−0.153	0.878
H10	NS	→	MT	−0.039	0.045	−0.723	0.470

## Data Availability

The data are available from the corresponding author upon reasonable request.

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
