# Peer review of "The Factors Influencing Users’ Trust in and Loyalty to Consumer-to-Consumer Secondhand Marketplace Platform"

_behavsci, 2023, doi:10.3390/bs13030242_

Round 1
Reviewer 1 Report
This study explores the mechanisms of trust in the C2C secondhand platform (Karot). More specifically, this manuscript examines how constrained-based and dedication-based mechanisms form two types of trust, including platform trust and mutual trust among users, which in turn affect loyalty. A major contribution of this study is to expand the knowledge on forming trust in C2C secondhand platforms. While the study is timely and interesting, it has several major deficiencies that need to be addressed. I have outlined some comments and suggestions for the authors' consideration in future revisions:
Abstract
In the abstract section, please clarify what precisely is the contribution of this study. For example, the author(s) may claim that, based on what they wrote, this manuscript contributes to confirming the significant role of two types of trust in forming loyalty.
Introduction
This manuscript seems to be a case study of Karot. It can reduce the contribution of this study. I think the introduction of this study needs to write the general context of the C2C secondhand marketplace platform. If not, this study should explain why this study focuses on the case of Karot.
There is a lot of Korean news article in this study. I think English-oriented readers cannot understand the references in this study. The study needs to update the news articles in this study to English versions.
This study focused on a Korean case. I think it is better to explain why the Korean case is important to understand a secondhand marketplace platform.
Research Background
The author(s) need to review the recent studies on C2C secondhand marketplace platforms. It is necessary to briefly explain how existing studies on users’ acceptance or post-adoption of the platforms have been conducted in the theoretical background section, how this manuscript differs from them, and what gaps are filled.
This manuscript only presents existing literature to develop hypotheses but does not explain how they can be applied in the context of C2C secondhand marketplace platforms. I think the authors need to apply existing literature to develop each hypothesis
It is hard to understand why loyalty is chosen as a dependent variable. According to chapter 2.2., the authors insisted that commitment is a predictor of loyalty, as well as dedication, which affects affective commitment. In other words, loyalty is the next level of commitment. I think there is a leap of logic to develop the hypothesis between trust and loyalty.
This manuscript does not fit neatly within the social exchange model in its current state. Many users in Karot interact with each other to decrease the risk of goods or negotiation. Thus, I think the information interaction is associated with economic benefits in the case of Karot.
The construct of no-show can be included in both constrained-based and dedication-based mechanisms. Thus, Figure 1 should be revised.
Furthermore, to develop H6, the authors cited the study of Palmatier et al. (i.e., “Palmatier et al. [45], when people perceive that they have similar lifestyle, status, culture, values, or goals with other people, they feel trust toward them”). It is hard to understand how individuals’ lifestyles, status, culture, values, or goals are related to psychological ownership.
Method
This study needs to suggest a correlation table among the constructs.
Discussion and Conclusion
I think it is the most interesting finding in this study - the opposite effect between mutual trust and loyalty. However, the study explained the opposite effect without any references. I think it will be a much better study if this phenomenon is explained based on the existing theory.
Reviewer 2 Report
This study has provided some interesting findings, however, the authors are recommended to amend the paper based on the following comments.
1. Introduction
- The authors are recommended to present the importance of consumer-to-consumer platform by using global statistics, for example, the global penetration rate, usage rate, time spent on consumer-consumer platforms.
- The authors are also recommended to present the knowledge gaps and highlight the theoretical and managerial implications.
2. Literature review
- The authors are recommended to highlight the importance of consumer-to-consumer interaction via social-media platforms. For example, Cheung et al. (2021) has found that consumer-to-consumer interaction is influential in evoking consumers' positive emotions, and subsequently drives their behavioural intentions.
Cheung, M. L., Pires, G. D., Rosenberger, P. J., Leung, W. K., & Sharipudin, M. N. S. (2021). The role of consumer-consumer interaction and consumer-brand interaction in driving consumer-brand engagement and behavioral intentions. Journal of retailing and consumer services, 61, 102574.
Abdul-Ghani, E., Hyde, K. F., & Marshall, R. (2019). Conceptualising engagement in a consumer-to-consumer context. Australasian marketing journal, 27(1), 2-13.
- The authors are also recommended to present the previous findings about emotional and informational interaction in driving trust and committment. For example, Cheung et al. (2022) found the importance of emotional and informational interactions via digital platforms in driving consumers' perceived value and subsequently strengthens their committment to the digital platforms.
Cheung, M. L., Leung, W. K., Cheah, J. H., & Ting, H. (2022). Exploring the effectiveness of emotional and rational user-generated contents in digital tourism platforms. Journal of Vacation Marketing, 28(2), 152-170.
The authors are recommended to amend the literature review section before the next submission.
Discussion and implications:
- The authors are recommended to present how the findings contribute to the literature in the area of consumers' interaction. Previous findings based on various theories should be presented and followed by the contribution of this study.
For example, based on SOR framework, Cheung et al. (2022) found the importance of emotional and informational interaction in drivng consumers' perceived value. Drawing upon service dominant logic, Cheung et al. (2021) found the importance of consumer-to-consumer interaction in evoking emotional attachment and behavioural intentions. Subsequently, the authors can present the unique contribution of this study.
Good luck, and all the best.
Reviewer 3 Report
Dear Authors,
thank you for giving me the opportunity to read your manuscript.
Overall, it is well written. However, I suggest to revise the manuscript in order to strengthen its contribution.
Please, clearify your research aim and the research motivation.
Please, highlight the contribution to the existing literature and the implications for practice of your results/study.
Please, check the language.
Reviewer 4 Report
Dear authors
My personal opinion is that after reading the paper, the manuscript is of potential interest to the readership of this journal, but there are issues that must be addressed:
In general:
1. Background – Expand a little more to highlight the research problem to highlight the study's need.
2. Methodology - expand a little more.
3. Findings: Should align with the study goal.
Introduction
The positioning of the paper is not entirely clear. The author is better to explain the gap in this article further.
A concise introduction to enable the reader's understanding of the research problem.
what research gap the paper aims to fill, what contribution the paper provides, and why the contribution is important.
Literature review
The paper should relate coherently and convincingly with issues of real-world significance. This is a crucial phase contributing to research design.
Suggestions
• Add more information to enable readers' understanding of the authors' view.
Methodology
The method should be adequately described to show how the research was conducted to improve clarity and transparency.
Findings and discussion
Needs clear and comprehensive explanations to assist readers' understanding.
Conclusion
The conclusion falls short of providing sufficient information that would allow a reader to understand the contribution of this research. What was found?
Reference.
- Using the following reference could be beneficial as these add more evidence to the literature review section:
Investigating social capital, trust and commitment in family business: Case of media firms. Journal of Family Business Management, 12(4), 938-958.
The effect of team performance on the internationalization of Digital Startups: the mediating role of entrepreneurship. Int. J. Hum. Capital Urban Manage, 8(1), 14.
Best of luck with the further development of the paper.
Round 2
Reviewer 1 Report
The author(s) did a great job revising the manuscript.
Reviewer 2 Report
The amended paper is acceptable. I recommend the editor to publish the paper in the present form. Thanks.
Reviewer 4 Report
Dear authors
Hope you are doing well. According to the review of this article, the corrections have been made.
Good luck